# Current Status and Challenges Associated with CNS-Targeted Gene Delivery across the BBB

**DOI:** 10.3390/pharmaceutics12121216

**Published:** 2020-12-15

**Authors:** Seigo Kimura, Hideyoshi Harashima

**Affiliations:** 1Laboratory for Molecular Design of Pharmaceutics, Faculty of Pharmaceutical Sciences, Hokkaido University, Sapporo 060-0812, Japan; harasima@pharm.hokudai.ac.jp; 2Laboratory of Innovative Nanomedicine, Faculty of Pharmaceutical Sciences, Hokkaido University, Sapporo 060-0812, Japan

**Keywords:** Gene therapy, BBB, AAV, non-viral vectors, non-invasive delivery, neurological disorders

## Abstract

The era of the aging society has arrived, and this is accompanied by an increase in the absolute numbers of patients with neurological disorders, such as Alzheimer’s disease (AD) and Parkinson’s disease (PD). Such neurological disorders are serious costly diseases that have a significant impact on society, both globally and socially. Gene therapy has great promise for the treatment of neurological disorders, but only a few gene therapy drugs are currently available. Delivery to the brain is the biggest hurdle in developing new drugs for the central nervous system (CNS) diseases and this is especially true in the case of gene delivery. Nanotechnologies such as viral and non-viral vectors allow efficient brain-targeted gene delivery systems to be created. The purpose of this review is to provide a comprehensive review of the current status of the development of successful drug delivery to the CNS for the treatment of CNS-related disorders especially by gene therapy. We mainly address three aspects of this situation: (1) blood-brain barrier (BBB) functions; (2) adeno-associated viral (AAV) vectors, currently the most advanced gene delivery vector; (3) non-viral brain targeting by non-invasive methods.

## 1. Introduction

The aging of society has now arrived, and this has been accompanied by an increase in the absolute numbers of patients of neurological disorders, such as Alzheimer’s disease (AD) and Parkinson’s disease (PD) and this increase is global in nature [1]. Neurological disorders show the highest disability-adjusted life years (DALYs) and are the second leading cause of death. Neurological disorders, especially AD and other forms of dementia, affect not only the patients themselves, but also the people around them such as their families and caregivers. As a result, such diseases are thought to have a larger social burden compared to other diseases. The most critical points in the current situation is that there is no effective treatment despite the fact that the number of patients increase with the aging of the population. Based on a questionnaire survey for physicians in Japan, one of the countries with the most rapidly aging population, neurological disorders are recognized as diseases/symptoms for which the development of novel treatment methods and drugs are urgently required [2]. According to a recent report, neurological disorders have low levels of both drug contribution and treatment satisfaction, and the development of new drugs would be highly desirable (Figure 1 and Figure 2).

The bottleneck in drug development for CNS diseases is the absence of effective drug delivery system (DDS) technology for delivering the therapeutic agents into the brain [3]. Less than 1% of the dose of most systemically administered compounds and drugs actually reach the brain. It may not be possible to efficiently deliver a drug to a human disease site, and the expected efficacy and safety may not be obtained, even if a candidate compound that would be expected to be effective at the non-clinical level can be identified. For example, it is thought that trastuzumab is effective for the treatment of a brain tumor since the brain tumor that has spread from a HER2-positive breast cancer into the brain remains HER2-positive. However, since the antibody trastuzumab cannot be efficiently delivered to the brain, it becomes difficult to actually use it for therapy [4]. Thus, even if there are drugs that would be expected to show theoretical effects, a DDS technology that meets those expectations does not currently exist, and the development of therapeutic agents for CNS may not progress. In other words, if a drug could be delivered to the brain using a DDS approach, it may have the potential to dramatically improve the prognosis of diseases in the brain.

Actually, there is a drug, (Spinraza), that is used for the clinical treatment of Spinal Muscular Atrophy (SMA) [5]. Spinraza is an antisense oligonucleotide (ASO)-based drug, which induces the expression of the functional survival motor neuron (SMN) protein from the SMN2 gene by exon inclusion. The efficacy and safety of the drug was confirmed in a randomized double-blind parallel-group comparison study of infants with SMA. However, it should be noted that the safe and reliable administration of this drug involves intrathecal administration. In the case of SMA in infants, many patients develop scoliosis due to disease progression, and it is necessary to consider the use of ultrasonic imaging for the correct intrathecal injection, which means that there are technical difficulties associated with this type of treatment [6]. Moreover, Spinraza needs to be administered every four months for infant SMA patients and every six months for non-infant SMA, and there are concerns regarding the risk for infections and tissue damage with each administration [7]. Given the complexity of this, highly invasive intrathecal administration, a DDS technology that can efficiently deliver drugs to the CNS disease site by a relatively low-invasive administration method such as intravenous injection would provide medical care with much less physical burden on the patients.

Gene therapy is currently a subject of great interest since it represents one of the most promising medicines for treating diseases that are currently incurable with conventional medicines [8,9,10]. In fact, gene therapy drug approval has been progressing at an increasing pace since 2012, as shown in Table 1. The concept of gene therapy originated in the 1970s, and in 1990, the first clinical trial involving the use of gene therapy was conducted for treating an adenosine deaminase (ADA) deficiency. However, the clinical trials failed to unequivocally show a clear efficacy, and in 1995, the Orkin-Motulsky Report from the National Institute of Health (NIH) appeared, noting the need to focus on basic research. The Gelsingner case in 1999 (death from adenovirus-induced systemic inflammatory response syndrome) and leukemia caused by a chromosomal insertion mutation of a retrovirus vector were reported, and gene therapy entered a period of stagnation. Since 2008, however, gene therapy development has been rekindled with the development of improved vectors, and gene therapy has now become a more active area of research [10]. Although most gene therapies involve local administration or ex vivo gene transfer (Table 1), the advent of Zolgensma confirmed that in vivo targeted gene therapy is a clear possibility and is expected to further accelerate the development of DDS technology in anticipation of gene therapy. In addition, gene therapy has emerged as having potential for the treatment of CNS diseases in the past decade [11,12,13,14,15]. Gene therapy strategies and therapeutic effects for CNS diseases that have been demonstrated are summarized in Table 2. The main gene therapy strategies involve the replacement of defective/dysfunctional genes and silencing mutant genes. In PD, which is characterized by loss of dopaminergic neurons in the substantia nigra and a decrease in dopamine levels in the striatum, the transfer of genes that encode for several enzymes such as aromatic l-amino acid decarboxylase (AADC) that converts l-dopa to dopamine [16], glutamic acid decarboxylase (GAD) that modulates production of the neurotransmitter GABA (γ-aminobutyric acid) are all possibilities [17]. In AD treatment, the main strategy is to remove extracellular amyloid since amyloid deposition is thought to be central event in AD [18]. It has been reported that an apolipoprotein E (APOE) isoform influences the risk for AD, and APOE4 is known to be a strong genetic risk factor [19]. APOE4 increases brain amyloid-β (Aβ) pathology relative to other APOE isoforms, and the absence of APOE is protective [20]. Other reports have also shown that APOE secreted by the glia stimulates neuronal Aβ production with an APOE4 > APOE3 > APOE2 potency rank order [21]. It has been reported that Aβ production is suppressed by attenuating β cleavage and promoting α cleavage by editing amyloid precursor protein (APP), which is a precursor protein of Aβ, with CRISPR-Cas9 [22]. The polyglutamine binding protein 1 (PQBP1) that is a major regulator of synapse-related proteins has been identified as an alternate target to amyloid deposition for AD treatment [23]. However, most of the gene vectors used in the above reports are administered by invasive local injection, and such brain targeting continues to be a challenge, although Zolgensma, an intravenously administered gene therapy for SMA, was approved in 2019 [24,25]. As mentioned above, an invasive local injection to CNS sites have a high risk. In addition, DDS technology is much more important in cases of nucleic acid-based medicines, such as gene therapy, due to their lower stability in the body compared to other conventional medicines, such as small molecule drugs and antibody drugs. Therefore, delivery to the brain is the biggest hurdle in developing new drugs for the treatment of CNS diseases, especially in the case of nucleic acid-based medicines.

The blood-brain barrier (BBB) is the biggest limiting hurdle to drug delivery to the brain in the case of the use of systemically injected vectors [3]. The BBB is a barrier that prevents several circulating molecules including harmful agents from entering the brain, and controlling the proper balance of nutrients. In terms of drug delivery to the brain, a rational strategy is taking advantage of innate BBB functions such as BBB receptors/transporters that promote substance transfer into the brain [40,41,42]. Several nanoparticles (NPs) designed for receptor mediated transcytosis have been reported; transferrin (Tf) receptor [43,44,45,46,47], nicotinic receptor [48,49,50], low density lipoprotein receptor (LDLR) [51,52,53,54,55], glucose transporter (GLUT) [56,57]. Although the usefulness of those receptor mediated strategies have been demonstrated, we need to consider BBB functions and pathologies because there are differences in BBB structures between the normal brain and the diseased brain [58,59,60]. Based on the above, an understanding of the mechanisms of BBB transport and BBB features in normal and pathological conditions is needed. Thus, we discuss BBB transporters, mechanisms of BBB crossing, and BBB breakdown in pathological conditions in this review.

Gene delivery vectors are mainly classified into two types, viral and non-viral vectors. Viral vectors have been demonstrated to have a 10 times to 1000 times higher efficiency of gene transfer in comparison to non-viral vectors [61,62,63]. Due to the high transfection efficiency of these vectors, viral vectors have been used in many clinical trials in gene therapy [8]. There are several types of viral vectors available for gene delivery [64], and among these, adeno-associated virus (AAV)-vectors have been widely used in many studies related to gene therapy [65,66,67] due to its features such as the potential of gene transfer to non-dividing/differentiated cells, long-term expression, relatively weak immunogenicity, and especially their applicability to in vivo gene delivery (tissue tropism) compared to other viral vectors. Additionally, AAV-vectors are particularly heavily used to transduce the therapeutic genes to CNS site for the treatment of neurodegenerative disorders [12,65]. Actually, the AAV9 serotype is used as a vector in gene therapy for the treatment of SMA, Zolgensma, as mentioned above. CNS-targeted AAV serotypes and Zolgensma are discussed in later sections. However, viral vectors have several demerits such as high immunogenicity, small carrying capacity of therapeutic genes, difficulties associated with scale-up preparation, and high costs. In terms of these points, non-viral vectors offer some advantages in spite of their low transfection efficiency compared to viral vectors [68]. Various types of non-viral vectors have been studied and developed in attempts to increase efficiency from the viewpoint of controlling biodistribution and the intracellular trafficking of the vectors [69,70,71,72,73,74]. Several synthetic vectors are available for gene delivery. One strategy depends on using therapeutic nucleic acids that are conjugated with different functional devices such as peptides, polymers, sugars, proteins, antibodies or aptamers [75]. Another strategy depends on encapsulating the nucleic acids in nanoparticles (NPs) in which the size can be tuned [76]. Conjugated systems are small in size and can easily be eliminated from the body by renal clearance. Furthermore, the nucleic acids in the conjugates are not protected and must be chemically modified to resist degradation in the circulation. On the other hand, NP systems are sufficiently large that renal clearance can be avoided and can provide more protection for the nucleic acids in the circulation. The NPs used for gene delivery can be broadly classified to polymeric and lipid nanoparticles (LNPs). In 2018, the first approved LNP-based RNA interference (RNAi) therapeutic, OnpattroTM (Patisiran) for the treatment of hereditary transthyretin-mediated amyloidosis (hTTAR amyloidosis), has opened the era of nucleic acid nanomedicines [77]. This breakthrough in the field of non-viral gene delivery received substantial interest worldwide and clearly pointed out the importance of non-viral systems such as LNPs for developing more approved drugs in the future. Additionally, there are also reports regarding non-viral brain-targeted systems, and we will introduce them and discuss the strategies for brain targeting focusing on systemically injected systems.

In this review, we provide a comprehensive insight into the developments of successful drug delivery to the CNS for the treatment of CNS disorders by gene therapy.

## 2. BBB Function

In this section, we briefly mention the general literature regarding BBB characteristics, because understanding the anatomical and physiological features and functions of the BBB is essential for designing rational brain-targeted systems. There are four parts in the section; (A) General structure of the BBB, in which we introduce the cell types that make up the BBB, (B) Transporters expressed on the BBB, mainly we focus on receptors and transporters that are likely to be used for nanoparticle delivery, (C) Mechanism of BBB crossing, this part outlines the BBB penetration mechanism reported in immune cells and AAV vectors, (D) BBB breakdown under pathological conditions, we also need to consider the differences in BBB structure and function between normal conditions and diseased conditions, we discuss how they are different.

### 2.1. General Structure of the BBB

The BBB is the interface that regulates transport between the circulation and neural tissue. The structural basis of the BBB is provided by the endothelial lining of the brain microvasculature. The brain microvascular endothelial cells are specialized cells that differ from peripheral capillaries: no fenestration, decreased pinocytosis and transcytosis, tight inter-endothelial junctions (TJs), high expression and asymmetric localization of transporters and high metabolic activity [78,79,80,81,82]. The development and maintenance of BBB function is dependent upon the concerted interaction of multiple cell types with the brain capillary endothelium, namely: astrocytes, pericytes, microglia and neurons (Figure 3). Astrocytes are star-shaped glial cells that constitute a large portion of brain cells and the glia is not homogenously distributed in the brain [83,84]. Their main functions are to supply neurons with nutrients and to regulate synaptic activities [85]. In addition, astrocytic end-feet cover almost the entire BBB endothelium [86], and modulate the availability of certain nutrients to the brain and the expression of the receptors/transporters that recognize these nutrients [79,87,88]. In addition, they can also change the BBB permeability through the modulation of TJs and efflux pumps [89,90]. Pericytes play an important role in stabilizing blood vessels and maintaining their function. They are present around micro blood vessels, and the ratio of vascular endothelial cells to pericytes differs depending on the specific organ (Ex. endothelial-to-pericyte ratio; human skeletal muscle (100:1), CNS (1:1~3:1) [86,91,92,93]). It has also been reported that pericyte density and coverage are correlated with endothelial barrier properties, endothelial cell turnover which means “large coverage = less turnover”, and orthostatic blood pressure [93,94]. Pericytes are able to modulate BBB integrity (increase of transendothelial electrical resistance (TEER)), transcytosis rate, and the expression of efflux pumps [95,96]. Microglia act as the resident immune cells of the brain, and play a role in the innate immune response in the CNS [97]. They have different morphologies depending on the activation state, and under different physiological conditions [98]. Microglia has a double-edged sword in brain pathologies, which means that they survey the influx of blood-born components into the CNS to protect the brain under normal condition, but, on the other hand, their dysregulation has been implicated in the initiation and progression of neurological disorders [98,99]. Moreover, microglia contribute to BBB integrity by exerting dual effects on BBB permeability, the initial protection of the BBB integrity but microglia develop into a phagocytic phenotype to increase BBB permeability as inflammation progresses [100]. These different cell types establish the BBB structure and function via interactions between each other.

### 2.2. Transporters Expressed on the BBB

Despite the fact that the BBB is a tight physiological barrier, vital molecules such as nutrients, neurotransmitters, and amino acids can cross the barrier to enter the brain. The specific receptors and transporters that are expressed on the BBB make this possible [58,101]. Brain endothelial transport systems regulate molecular exchanges between blood-and-brain and brain-and-blood. These BBB receptors/transporters have been validated by transcriptomics and/or proteomics [95,102,103,104,105,106,107,108,109,110,111,112]. Rational brain targeted DDS is based on the understanding of drug and/or carrier interactions with those BBB receptors/transporters. As shown in Figure 4, the main internalization pathways carried on by receptors/transporters that are expressed on the BBB include transporter-mediated pathway, clathrin-dependent pathway, and caveolae-dependent pathway. Once internalized, depending on the internalization pathway and/or the type of vesicles, it undergoes different intracellular routes, such that it can interact with other cellular compartments or vesicles such as endosomes, lysosomes and exocytose. This section reviews representative receptors/transporters in the above three internalization pathways, and discusses the pros and cons of each in terms of BBB crossing.

First, we introduce glucose transporters (GLUTs) as a representative of transporter-mediated transcytosis. GLUTs play a role of the transport of glucose and/or other hexose/pentose sugars from the blood into the cell. GLUTs are expressed throughout the body, and there are various GLUT family members that have variable expression levels depending on the site where they are located. The most abundant GLUT is GLUT1, which is expressed in red blood cells and endothelial cells with a high glycosylation rate (55 kDa), and in astrocytes, neurons and microglia with a low glycosylation rate (45 kDa). GLUT3 is the main transporter expressed on neurons. GLUT5, the major GLUT in microglia, transports fructose and has a low affinity for glucose [113]. GLUTs are expressed at both the luminal and abluminal sides of BBB endothelial cells for regulating brain glucose levels. GLUTs are transporters and not receptors, so the internalization pathway is a process that involves conformation change after ligand binding, and the release of the ligand into the cell (Figure 4). However, this process may differ in the case of nanoparticles due to their nanometric size. It appears that the particles attach to the transporter and enter the cell with the recycling of the transporter. In hypoglycemic animals, the total GLUT1 of micro-vessel protein and luminal GLUT1 are increased [114]. GLUT1 on the cell membrane undergoes endocytosis and is pooled in intracellular vesicles [115,116]. The above reports suggest that the dynamic intracellular recycling of GLUT1 occurs in response to changes in blood glucose level. Actually, it has been reported that the recycling of GLUT1 improves the translocation of nanocarriers to the brain [56]. The authors used glycaemic control in order to boost the crossing of glucosylated nanocarriers of the BBB into the brain. As mentioned above, some strategies such as the use of recycling of transporters should be considered for allow nanoparticles to cross the BBB through transporter-mediated pathways.

Second, we describe the transferrin receptor (TfR) as a representative of clathrin-dependent transcytosis. TfR is one of the most extensively studied and exploited receptor that is expressed on the BBB [117]. A high expression level of TfR at the BBB makes them one of the most desirable receptors for drug delivery across the BBB [43,44,45,47,117,118], although it is also expressed in other tissues throughout the body [119]. Two types of TfR have been reported, TfR1 and TfR2 [120]. TfR2 is mostly expressed by erythroid cell lines, normal erythroid cells at various stages of differentiation, and leukemia and pre-leukemia cells [121]. Although the differences between TfR1 and TfR2 are not fully understood, it has been reported that the TfR-mediated transferrin-bound iron uptake is mediated primarily via TfR1 but not TfR2 in HuH7 human hepatoma cells [122]. In addition, it was recently reported that TfR2 facilitates iron transport from lysosomes to mitochondria in erythroblasts and dopaminergic neurons, and defects in the TfR2 can cause systemic iron overload, hemochromatosis [123]. TfR1 is highly expressed at the luminal side of the BBB, and promotes the uptake of iron by the brain by binding to transferrin saturated with iron [124]. This process involves receptor-mediated transcytosis (RMT) through clathrin-dependent transcytosis and involves the formation of an endosome (Figure 4). After acidification of the lumen of the endosome, the binding affinity between the TfR complex and iron become weaker, resulting in its dissociation. The transferrin and TfR are then recycled back to the cell surface [125]. There are several limitations of TfR1 as a targeted receptor for crossing the BBB: (1) its expression throughout the whole body can lead to non-specific targeting; (2) competition between endogenous transferrin and the targeting ligand that binds to TfR1 can reduce targeting efficiency [126]; (3) the inability of TfR transcytosis to occur through brain capillary endothelial cells to the abluminal membrane and brain parenchyma [127,128]. In a recent report [128], the use of TfR-targeting was found to increase the binding between immuno-liposomes and brain capillary endothelial cells, however, the transcytosis of immuno-liposomes was not evident. Namely, such a targeting approach might increase the uptake of nanoparticles at the BBB several fold but might not lead to BBB transport. In order to overcome the above limitations, the use of monoclonal antibodies (mAbs) has been suggested [128,129,130]. One such example is OX26 mAb [119,128]. OX26, a mAb for the specific targeting of rat TfR, binds to an allosteric site on TfR that is distinct from the transferrin binding site, and does not interfere with Tf binding [131,132].

Third, we use a low-density lipoprotein receptor (LDLR) as an example of caveolae-dependent transcytosis. The LDLR at the BBB plays a role in controlling the homeostasis of cholesterol, thus acting as a mediator for the internalization of cholesterol-rich low-density lipoproteins (LDL), apolipoprotein B and E. The process occurs at the BBB, although it is significantly greater at the liver [133]. Recognizing apolipoproteins promotes endocytosis via a caveolae-dependent mechanism. This pathway is different from the clathrin-dependent pathway due to the fact that caveosomes do not deliver their cargo to lysosomes [134,135]. This feature is advantageous in terms of the intracellular trafficking of therapeutic genes. There are two main strategies for targeting LDLR; (1) protein corona-mediated targeting; (2) ligand-based targeting. Once nanoparticles are injected into the systemic circulation, nanoparticles encounter serum components, such as proteins, resulting in the formation of a protein corona on the surface. The formation of a protein corona is critical for the design of efficient and safe nanoparticles for tissue-targeting, nanomedicines, and other applications, so research related to the protein corona is a subject of great interest [136,137,138,139,140,141,142,143,144,145,146]. Although protein corona formation on a nanoparticle surface may adversely affect targeting [147], controlling them can also be applied to achieve more effective targeting [51,54,55,148,149,150,151,152,153,154]. With regard to the brain targeting through LDLR-mediated transcytosis, it was proposed that the use of certain apolipoproteins, such as ApoE and ApoA1, would be useful as endogenous ligands for crossing the BBB. One such example is polysorbate-80 coated nanoparticles which bind to ApoE and interact with the LDLR, resulting in an increased brain accumulation compared to nanoparticles without ApoE binding or with low levels of ApoE binding [51,155,156,157]. Below, we also describe the brain targeting strategies that involve a protein corona. Angiopeps, which were derived from the Kunitz domains of aprotinin and other human proteins [158], is known as the most extensively explored ligand for targeting LDLRs for BBB crossing. Angiopep-2, one of the family of angiopeps, exhibited a particularly higher transcytosis capacity and parenchymal accumulation compared to transferrin, lactoferrin, and avidin, of which BBB transport is mediated by low-density lipoprotein receptor-related protein-1 (LRP1) [159]. In fact, angiopep-2 could act as a ligand for LRP in the delivery of drug-loaded nanoparticles [158,159,160,161,162,163,164]. The use of angiopep-2-conjugated PEG-PCL nanoparticles showed LRP-mediated transcytosis and the system could target glioma cells in vivo after intravenous injection [164]. A biodistribution study using nanoparticles labeled with near-infrared fluorophore also showed that angiopep-2 functionalized nanoparticles were mainly localized in the striatum, hippocampus, and cerebellum in the brain [160]. Furthermore, angiopep-conjugated dendrigraft poly-L-lysine nanoparticles were used for gene delivery to produce neuroprotective effects in a chronic PD model [163]. In that report, angiopep-conjugated nanoparticles exhibited a higher uptake and gene expression in the brain compared to unmodified nanoparticles, and nanoparticles containing human glial cell line-derived neurotrophic factor gene showed an improved locomotor activity and the apparent recovery of dopaminergic neurons compared to other control groups. Although not a gene delivery sytem, angiopep-2 has been used in clinical trials for the delivery of small molecule drugs to the brain for the treatment of brain metastases (BM) [165,166,167]. ANG1005, consists of three paclitaxel molecules that are covalently linked to angiopep-2, and was designed to cross the BBB and blood-cerebrospinal barriers and to penetrate malignant cells via the LRP1 transport system. In a phase I study, ANG1005 was detected in recurrent glioma tumors that had been resected 3 to 6 h after a single intravenous administration of ANG1005, providing evidence of transport across the BBB and tumor penetration, and an antitumor effect was observed in both CNS and peripheral tissue at does ranging from 420 to 650 mg/m^2^ in patients with BM [166,167]. In a phase II study, adults with recurrent brain metastases from breast cancer, with or without leptomeningeal carcinomatosis were administered ANG1005 intravenously at a dose if 600 mg/m^2^, resulting in notable CNS antitumor activity and demonstrated good efficacy systemically [165]. A randomized phase III study of ANG1005 is underway to further evaluate the treatment effect seen in patients with leptomeningeal carcinomatosis.

In this above section, we described several transport systems for BBB crossing and some examples of their use in drug delivery, especially nanoparticle-based delivery. While we focused on the three pathways, namely: (1) transporter-mediated transcytosis; (2) clathrin-dependent transcytosis, and (3) caveolae-dependent transcytosis, in terms of nanoparticle-based delivery, it is thought that caveolae-dependent pathways, such as the LDLR-related pathway, may be a more suitable pathway among these from the viewpoint of intracellular trafficking.

### 2.3. Mechanism of BBB Crossing (Immune Cells, AAV Vector)

It is highly difficult to deliver macromolecules into the brain, as mentioned above. One of the rational strategies for achieving this is the use of endogenous transport systems, such as transporters/receptors on the BBB, examples of which were given above. Another useful strategy includes designing DDS carriers based on the migration mechanism of immune cells that migrate into the brain and the mechanism of viral vectors that are currently the most advanced vector for gene therapy. Thus, in this section, we summarize several reported mechanisms, namely: (1) the mechanism of immune cell migration into the brain and (2) endogenous factor required for CNS migration of AAV-vectors.

The recruitment of immune cells from the circulation is one of the most dynamic cellular responses to tissue damage and inflammation. Lymphocyte extravasation is regulated by selective interactions between lymphocytes and endothelial cells [168], which can show specificity in relation to the tissue site and/or organ [169]. However, classical thinking indicates that there seems to be no routine immune surveillance of the CNS due to barriers such as BBB. In fact, the CNS is an immune-privileged site, since it is well known that tissue grafts survive well when implanted into the CNS parenchyma [170]. On the other hand, activated lymphocytes can enter the CNS for immune surveillance [171,172,173,174]. Studies of the lymphocyte transfer capacity of the barriers (BBB and blood-cerebrospinal fluid barrier (BCSFB)) revealed that the BBB was breached and lymphocyte populations migrated across the endothelium to accumulate at sites of an antigen challenge during inflammatory processes in the CNS [171]. Three potential immune cell entry sites into the CNS are proposed, and these sites are localized to superficial leptomeningeal vessels, parenchymal vessels, and the choroid plexus [174]. The activation of endothelial cells and associated cells such as astrocytes, which occurs on the development of a CNS injury, lead to the reduced integrity of tight junctions, facilitating the migration of leukocytes across the BBB into the brain [175,176]. While molecular pathways for leukocyte entry into the CNS are different depending on the inflammatory stimulus and the CNS compartment affected, adhesion molecules are involved in the migration of leukocytes [175,177]. Inflammation in the CNS leads to an increased expression of adhesion molecules on endothelial cells of the BBB, such as the intercellular adhesion molecule-1 (ICAM-1), vascular cell adhesion molecule-1 (VCAM-1), and the platelet endothelial cell adhesion molecule (PECAM1) [178]. These adhesion molecules play important roles in multi-step immune cell trafficking across the BBB, such as capture, rolling, adhesion and diapedesis. It has been reported that a lack of endothelial ICAM-1 and ICAM-2 in a mouse model of the BBB abrogates Th1 cell polarization and crawling [179], and it has also been demonstrated that the cell surface levels of endothelial ICAM-1 rather than BBB integrity influence the pathway of T-cell diapedesis across the BBB [180]. Interestingly, inflammatory conditions with high levels of endothelial ICAM-1 promote the rapid initiation of transcellular diapedesis of T-cells across the BBB, while intermediate levels of endothelial ICAM-1 favor paracellular T-cell diapedesis [180]. Furthermore, in the case of neutrophil diapedesis across the inflamed BBB, it has been shown that β2 integrin-mediated crawling on endothelial ICAM-1 and ICAM-2 is a prerequisite for transcellular diapedesis [181]. Considering the above, this mode of ICAM-1-mediated transport may be important under certain types of pathological conditions and relevant for drug delivery exploitation. Actually, it has been reported that nanocarriers modified on the surface by anti-ICAM-1 or VCAM-1 showed a higher uptake in an inflamed brain compared to non-targeted nanocarriers [182,183,184,185]. Other reports have proposed “the gateway theory” [186,187,188,189,190,191,192,193]. According to these reports, there is a gateway by which pathogens or immune cells enter the CNS, and the mechanism of gateway formation is a massive chemokine-inducing “inflammation amplifier” through the co-activation of the NF-kB and STAT3 pathways [187,191]. In that report [187], the activation of sensory nerves by artificially weak electrical stimulation of the quadriceps or triceps muscles increased the expression of chemokines and formed vascular gates for immune cells to enter the CNS in the dorsal vessels of the third lumber spinal cord or fifth cervical spinal cord to the fifth thoracic spinal cord, respectively. The location of gateways also depends on regional neural activation [190]. Those results reveal that a gate for CNS entry of immune cells can be controlled by the artificial regional neural activation, and controlling local neural stimulation represents a potential therapeutic strategy for treating inflammatory conditions in the CNS. While there are still many unknowns concerning this and further research is needed, it is possible that the strategic of the engineering biological state by the gateway control could be applied to deliver drugs into the CNS.

The AAV-vector is currently known as the most advanced gene delivery vector for CNS-targeted gene therapy. In a later part of this report, we summarize the CNS-targeted AAV-vector in detail. In this section, we introduce the reported mechanism for the crossing of the AAV-vector thorough the BBB. Recently, host cell factors involved in the gene transfer of AAV-vectors have been identified by a variety of comprehensive screenings and analyses [194]. AAV-PHP.B is the most efficient CNS-targeted gene delivery vector in rodents [195]. However, the AAV-PHP.B cannot show CNS tropism in some mouse strains and animal species [196,197]. Several research groups used this phenomena and identified a specific haplotype of the lymphocyte antigen 6 complex, locus A (Ly6a) (stem cell antigen-1 [Sca-1]) as the factor required for the BBB crossing of AAV-PHP.B [198,199,200]. Ly6a molecules are expressed on the surface of the BBB and they have several single nucleotide polymorphisms (SNPs). The results of AAV-PHP.B transduction with/without Ly6a showed that Ly6a facilitates binding and transduction both in vitro and in vivo [198]. Although Ly6a has been identified as an essential factor for the CNS tropism of AAV-PHP.B, primates contain no direct Ly6a homolog. The question therefore arises as to how we utilize this finding to apply gene delivery vectors for humans. It is thought that other cellular factors that share key properties with Ly6a may be prime molecular targets for gene delivery vectors in mice, non-human primates (NHPs), and humans.

### 2.4. BBB Breakdown under Pathological Conditions

The BBB is a highly tight barrier that limits the influx of substances into the brain. However, it has been reported that BBB disruption occurs under various pathological conditions of diseases such as strokes, traumatic brain injuries, MS, PD, AD, and other brain trauma [58,59]. In sporadic AD, the BBB is structurally and functionally disrupted. For example, the down-regulation of GLUT1 and LDLR, decreased the expression and functionality of ATP-binding cassettes (ABC) transporters [42,201]. Other features of BBB breakdown are summarized in Table 3. Disease-initiated BBB breakdown might present an opportunity for delivering therapeutic agents. Actually, in the pathological condition of ischemia, it is possible for small-sized liposomes to enter into the brain tissue [202]. However, vascular changes in BBB disruption include endothelial degeneration, reduced expression of TJs at the BBB, increased endothelial bulk flow transcytosis, disrupted BBB transporter expression, inflammation, and immune response, can all hinder the delivery of therapeutic agents to the brain. Furthermore, decreased function of carrier-mediated transport (CMT) and receptor-mediated transport (RMT) systems in neurodegenerative diseases complicates their use for therapeutic drug delivery [59]. Given this, we should consider the differences between normal conditions and pathological conditions in both the structures and functions of the BBB for designing an efficient DDS.

## 3. AAV Vector; Currently the Most Advanced Gene Delivery Vector

AAV-vectors are currently known as the most advanced gene delivery vector for crossing the BBB, and they are becoming a platform for the treatment of various neurological diseases [12]. AAV-vectors are used to transduce therapeutic genes to the CNS site for treatment of neurodegenerative disorders [12,65], and many serotypes have been developed by capsid modification strategies, such as directed evolution (Random mutation insertion, DNA shuffling, Peptide display) [214] and rational design [215], as shown in Table 4. Actually, the AAV9 serotype is used as a vector for gene therapy for the treatment of SMA, Zolgensma. In this section, we present some examples of AAV-vectors, especially brain-targeted systems, and summarize the currently-approved gene therapy, Zolgensma for the treatment of SMA. Finally, we briefly outline the challenges to using the AAV-vector at the moment.

### 3.1. Brain-Targeted AAV Vectors Developed So Far

AAV-vectors, especially AAV9 serotypes, are used for CNS-targeted gene delivery vectors that are administered non-invasively [216,220,221,222]. By capsid modification strategies, many serotypes of AAV-vectors have been developed in order to create far more efficient CNS-targeted vectors. AAV9 is the first identified serotype that has the ability to bypass the BBB [216,222]. However, the AAV9 serotype has several limitations in terms of more gene expression in astrocytes compared to neurons in adult animal brains, and the DNA package size. To overcome these limitations, tyrosine-mutant AAV9/3 (AAV.GTX) vectors have been developed [217]. A mutation can enhance gene expression efficiency through the inhibition of intracellular ubiquitination, in addition, the usage of neuron-specific promoters achieved selective transduction of neurons. Several cell type specific/selective AAV-vectors have been reported (brain microvascular endothelial cells [223], astrocytes [224]). In addition, an AAV that targets a specific subpopulation of astrocytes was developed using a specific promoter expressed in the target astrocyte subpopulation [225]. Multiplexed-Cre-recombination-based AAV targeted evolution (M-CREATE) was recently developed as an efficient methodology for identifying variants of interest in a given selection landscape [226]. The M-CREATE could identify capsid variants that can target distinct brain cell types. AAV-PHP.B has been identified by a capsid selection method, referred to as Cre-recombination-based AAV targeted evolution (CREATE), and the AAV-PHP.B shows at least a 40-fold greater transduction efficiency in the CNS compared to conventional standard, AAV9 [195]. As mentioned above, AAV-PHP.B failed to show CNS tropism in some mouse strains and animal species [196,197], and a specific haplotype of the lymphocyte antigen 6 complex, locus A (Ly6a) (stem cell antigen-1 [Sca-1]) has been identified as the factor required for AAV-PHP.B to cross the BBB [198,199,200]. AAV.B1 has been isolated as a novel CNS tropic AAV capsid after a single round of in vivo selection from a DNA shuffled AAV viral library [218]. The AAV.B1 showed a higher efficiency than AAV9 for gene delivery to the mouse brain, together with reduced sensitivity to neutralization by antibodies. AAV.AS, generated by the insertion of a poly-alanine peptide, showed 6- and 15-fold more efficient gene transduction than AAV9 after systemic administration in the spinal cord and cerebrum, respectively [219]. The AAV.AS transduced particularly high in the striatum (36% of striatal neurons). We summarized the AAV serotypes described above in Table 4.

### 3.2. Zolgensma; AAV9-Based Gene Therapy to Treat SMA

In May 2019, the FDA approved a gene therapy for SMA, Zolgensma [25]. It was developed by AveXis, a Novartis company, for the treatment of paediatric patients aged <2 years with SMA and bi-allelic mutations in the primary gene encoding the survival motor neuron protein, namely the SMN1 gene. SMA is a rare, progressive genetic disease, yet it is the number one genetic cause of infant death [227,228]. The genetic root of SMA is the SMN1 gene that is missing or not functioning properly. If this gene is missing or not functioning properly, the body cannot produce enough SMN protein, which is required for motor neuron survival. There is a backup gene for the SMN1 gene, called the SMN2 gene. However, the SMN2 gene produces only 10% of the working protein compared to the protein produced by the SMN1 gene. Some people can have more copies of the SMN2 gene and others can have fewer, so there is a wide range of severity in individuals who are affected by SMA, which means that the higher the severity, the fewer backup genes. Zolgensma delivers a functional SMN gene to neurons in order to improve the survival of motor neurons [24]. For SMA treatment, there is an antisense nucleic acid-based drug, called Nusinersen [5] which enhances the production of functional SMN proteins by a exon-inclusion mechanism. Both Zolgensma and Nusinersen demonstrated meaningful improvements in efficacy in clinical trials. There is a report that compares Zolgensma with Nusinersen with regard to the efficacy such as overall survival, event-free survival, improvement in motor function, and motor milestone achievements [229]. The results of Zolgensma (AVXS-101-CL-101; NCT02122952) and Nusinersen (ENDEAR; NCT02193074) clinical trials were indirectly compared in the report. The comparison suggested that Zolgensma may have an advantage relative to Nusinersen in the above items.

### 3.3. Controversy Concerning the Use of AAV Vectors

AAV-vectors are of particular interest as brain-targeted gene delivery vectors, and they are also widely used clinically. However, that does not mean there are no challenges. In fact, there is controversy about Zolgensma. Although significant efficacy was reported, there was a death in the Phase3 clinical study and detailed findings at necropsy remain unclear. The dose set by Zolgensma appears to be quite high (1.1 × 10^14^ vector genome (vg)/kg body weight), and this dose could cause organ damage, an immune response to the viral vector because it is reported that severe toxicities, such as liver damage, and the degeneration of sensory neurons, were observed in NHPs and piglets at 2 × 10^14^ vg/kg body weight [230]. Actually, the document says “Zolgensma could result in elevated liver enzymes and cause acute serious liver injury. An oral corticosteroid should be started the day before infusion with Zolgensma”. Furthermore, data manipulation was noticed in pre-clinical studies after FDA approval, although FDA has confirmed that it will continue to permit it to sell as it has no impact on safety and the efficacy of the drug [231]. There are also some remaining problems and challenges in terms of the use of the AAV-vectors such as problems associated with tissue-tropism preservation by animal species, neutralization antibodies against AAV, biodistribution after systemic administration, and high dose/large-scale production (cost problem). In this section, we discuss the above two problems, first, problems associated with tissue-tropism preservation by animal species and, second, neutralization antibodies against AAV.

Although some AAV-vector serotypes can produce efficient CNS tropism, there are differences in the pattern of CNS biodistribution in NHPs compared with mice. AAV-PHP.B has recently been shown to transduce the brains of mice with a higher efficiency compared with its parent serotype, AAV9, as mentioned above. Although AAV-PHP.B showed widespread and largely equal CNS transduction in mice following different injection strategies including two intravascular (intra-jugular vein and intra-carotid artery) and two intra-CSF routes (intra-cisterna magna and intra-lateral ventricle), a differential pattern of transduction in macaques was observed with broad cortical and spinal cord transduction being observed after intrathecal administration and only very low transduction following intravascular injection [197]. How do we apply the AAV-vectors to human use if there are differences in tissue-tropism depending on animal species? One solution is to devise a route of administration of the AAV-vectors such as is described in reference [197], but it is not desirable if the suitable route of administration is invasive method.

Since AAV infections occur naturally in humans, some patients can have pre-existing immunity. AAV was actively studied in the 1960s and 1970s, and AAV1–4 serotypes were identified at that time, and the positive rate of each neutralizing antibody was measured [232]. The results showed that the majority of the subjects are positive. Other reports also examined the prevalence of neutralizing antibodies against AAV serotypes [233,234,235]. The findings suggested that the destruction of transduced hepatocytes by cell-mediated immunity targeting antigens of the AAV capsid caused both the decline in gene expression and the transient transaminitis [236]. Gene expression was observed in monkeys with AAV8-neutralizing antibody-negative individuals, but not in positive individuals [237], indicating that neutralizing antibodies greatly affect the therapeutic effect. The neutralizing antibody positive rate tended to increase with increasing age [233]. Initially, the AAV-vector was administered by the direct injection into the target tissue, but was subsequently replaced by transvascular administration [238], which makes it more susceptible to neutralizing antibodies. Considering the above, the issue of whether a patient has neutralizing antibodies against AAV before the treatment usage should be demonstrated, and it appears that it would be more difficult to use AAV for older patients compared to younger patients.

## 4. Non-Viral Brain Targeting by Non-Invasive Methods

Viral vectors have a significantly higher transfection efficiency in comparison to non-viral vectors [61,62,63]. However, as mentioned above, there are several problems and challenges to the use of viral vectors with regard to their efficacy and safety profiles. In terms of those points, non-viral vectors offer some advantages, a safer and more flexible route to gene delivery [68,69,70], even in brain targeting [239,240,241]. As shown in Figure 5, there are several non-viral strategies for brain targeting using non-invasive methods. The purpose of this section is to present non-viral methods and focus on systemically injected non-viral systems for brain targeting as follows; (1) Active targeting using ligands/peptides-modification; (2) Protein corona (endogenous ligands); (3) Transient BBB disruption.

### 4.1. Active Targeting Using Ligands/Peptides-Modification

Most studies of drug delivery to the brain use active targeting approaches in order to produce a high brain-selectivity and reduce possible side effects. An active targeting strategy is a simple concept in which specific receptors/transporters present on the brain cells are used for targeting. This section provides an overview of reports on brain targeting with targeted-ligands, especially in macromolecule delivery such as proteins and nucleic acids.

One elegant and non-invasive strategy is to utilize peptides derived from neurotropic viruses or organisms as the targeting ligand. Peptide derivatives of the rabies virus glycoprotein (RVG) have been exploited as delivery ligands to target the brain [242,243]. The reason for using RVG for brain targeting can be explained by the features associated with rabies virus infections. The virus is bullet-shaped, with a length of 200 nm and a diameter of 80 nm [244]. The virion carries the RVG on the surface, which is responsible for cellular entry and virus fusion [245]. Based on the above facts, RVG would be a candidate for the use in brain-targeting. However, the rabies virus usually utilizes retrograde transport within the neuronal network to spread from the site of infection to the CNS, and the issue of whether the virus can penetrate the BBB is not known with certainty [246,247]. It is thought that peptide derivatives of RVG use different pathways to gain access to the brain as opposed to rabies virus. It has been documented that RVG mediates the delivery of small interfering RNA (siRNA) to the brain [48]. In that report, a chimeric peptide was synthesized by adding arginine residues at the carboxy terminus of RVG to enable siRNA binding. This RVG-9R delivered siRNA to neuronal cells after intravenous injection into mice, resulting in specific gene silencing in the brain. The proposed targeting mechanism suggests that the peptides and their conjugates specifically bind to nicotinic acetylcholine receptors on the BBB, and then penetrate the BBB by receptor-mediated transcytosis. Applications of RVG have also been explored for delivering therapeutic proteins and plasmid DNA (pDNA) to treat CNS diseases. Peptides derived from the RVG (RDP) delivered protein/plasmid-brain-derived neurotrophic factor (BDNF) into the brain and showed a therapeutic effect on stroke/PD model mice [248,249]. Furthermore, the RVG was used for brain targeted genome editing in adult mice by the injection of RVG-Cre into Cre reporter mouse lines via the tail vein [250].

As mentioned above, GLUT1 is often used as a target molecule for brain drug delivery since it is abundantly expressed on the BBB. However, in the case of nanoparticles for delivering macromolecules such as proteins and nucleic acids, it appears to need to take advantage of transporter recycling instead of the usual transport pathway of GLUT1. There are reports on brain delivery via GLUT1 by utilizing recycling [56,57]. In these reports, not only the precisely controlled glucose density on the surface of the nanocarrier but also glycemic control as an external trigger showed a dramatically enhanced brain accumulation of the carrier (>6% injected dose/g of brain), resulting in significant knockdown of a target RNA in various brain regions by an antisense oligonucleotide that was encapsulated in the carrier.

### 4.2. Protein Corona (Endogenous Ligands)

Despite an abundance of preclinical studies on nanomedicines, its clinical translation is still limited. One of the possible factors may be the formation of a “protein corona” on nanoparticles. It is widely accepted that once nanoparticles are systemically administered, they encounter serum components, such as proteins, resulting in the formation of a protein corona on the surface [137,138,141,146,251,252,253,254]. Protein corona formation is a double-edged sword, which means controlling them can be applied to more effective targeting [51,54,55,148,149,150,151,152,153,154] although protein corona formation on the nanoparticle surface may adversely affect the targeting [147]. With regard to brain targeting, ApoE is a well-known endogenous ligand, and brain delivery using nanoparticles with suitable affinity for ApoE has been conducted. As mentioned above, polysorbate-80 coated nanoparticles can bind to ApoE and enter the brain via a LDLR-mediated pathway [51,155,156,157]. There appears to be two modification modalities for controlling the protein corona, pre-coating before injection and controlled corona formation in the circulation. As an example of the former, it was reported that pre-ApoE4 decorated nanoparticles showed an improved brain accumulation compared to un-decorated nanoparticles [51]. However, random protein adsorption can cause access to the receptor-binding pocket to be inhibited [255,256]. It is thought that this can be avoided if the interaction mode of protein adsorption on nanocarrier-surface can be controlled. The latter example is that precisely manipulating the binding pattern of brain targeted apolipoproteins on the nanoparticle surface resulted in superior brain accumulation and showed a therapeutic efficacy in a glioma tumor model [54]. In the report, the liposomal surface was modified with a short nontoxic peptide derived from β-amyloid (Aβ_1–42_), which specifically interacts with the lipid-binding domain of apolipoproteins, for manipulating the modes of apolipoprotein adsorption. The liposomal system developed in this study led to the association of brain-targeting proteins in the circulation, and the receptor binding domains of these proteins were appropriately exposed on liposomal surface in the blood stream. The modification of brain targeting plasma proteins in the circulation has the potential to overcome the following drawbacks to the use of protein-modifications for clinical translation [257,258]: (1) protein modification makes production, storage, and transportation costly; (2) elevating immunogenicity is a risk; (3) the endogenous brain targeting protein competes with the receptor binding of the carriers.

### 4.3. Transient BBB Disruption

Finally, we briefly introduce a transient BBB opening strategies by using external stimuli (micro-bubbles (MBs) and focused ultrasound (FUS)) and others (chemical methods). Transient drug or gene delivery to the brain is possible by FUS irradiation of the brain following the intravenous administration of MBs [259,260,261,262,263]. FUS irradiation of a targeted region of the brain and the presence of MBs in the blood cause the transient disruption of the BBB, leading to promoting the permeation of nanoparticle through the BBB. Several reports demonstrated that FUS and MBs can be utilized to deliver therapeutic genes for curing neurodegenerative disease, especially in a PD mouse model [264,265,266,267,268,269]. There is strong evidence to show that neurotrophic factor can promote the regeneration of dopaminergic neurons to releive the PD syndrome [270,271,272]. The glial cell line-derived neurotrophic factor (GDNF) is one of the potent agents for PD treatment because of its neuroprotective and neurotrophic effects [272,273]. FUS and MBs increased brain gene transfection and dopamine levels were restored by FUS-triggered MBs GDNF gene delivered treatment, which produced an abnormal rotation reduction in PD mice [267] and rats [265]. There are several methods available for transiently disrupting the BBB in addition to FUS and MBs. One of the strategies involves the use of chemical compounds, such as Borneol [274,275], mannitol [276] and an adenosine receptor agonist [277,278,279]. Borneol, which is widely used as messenger drug in traditional Chinese medicine, can enhance drug permeation through various membranes including the BBB [274]. A recent report suggests that elevated levels of the expression of the intracellular cell adhesion molecule-1 (ICAM-1) contributes to Borneol gaining access to the BBB, although the detailed mechanism is unknown [275]. Mannitol can also cause chemical destruction resulting in the opening of the BBB by changing the osmotic pressure [276]. Adenosine receptors have been considered to be therapeutic targets in many diseases including CNS disorders [280], and it has been reported that the activation of adenosine receptors could elevate the permeability of the BBB in vivo [279]. In that report, lexiscan, an agonist of A2A adenosine receptors, -conjugated dendrimers (nanoagonists) improved the accumulation of a model drug in the brains of mice that had been pretreated with nanoagonists compared to dendrimers without lexiscan. It will be important to verify the risk of foreign invasion into the brain through the opening of intracellular gaps. Furthermore, it should be noted here that it is necessary to adjust this so as not to cause chronic BBB disruption since chronic disruption is injurious to the brain and induces neurodegeneration [281]. Considering the above, it may be better to use the transient BBB disruption method for the treatment of regional diseases such as brain cancer.

The above section highlights three mainly used non-viral strategies focused on systemically injected non-invasive methods. There are, however, other non-invasive methods such as intranasal delivery that are beyond the scope of this report, but have also been efficient in allowing the delivery of therapeutic agents [282]. Briefly, intranasal delivery is a non-invasive method, known as the nose-to-brain route, for delivering various therapeutic molecules including small molecules [283], proteins [284,285], and oligonucleotides as well [286] and gene delivery vectors [287,288]. There are two proposed pathways for intranasal delivery bypassing the BBB: the olfactory pathway and the trigeminal pathway. In the former, agents are distributed in the olfactory bulb through axonal transport by their nerves or passage through the nasal epithelium. In the latter, agents are distributed in the hypothalamus and brainstem. There are still obstacles to overcome, such as the mucus layer and the epithelium, where efflux transporters and several metabolizing enzymes are located. Other limitations to the intranasal delivery include the restricted volume of administration. For more information about other options, the readers are referred to other reviews and reports [282,289,290,291,292,293].

## 5. Summary and Future Directions

The development of new drugs for the treatment of neurological disorders, such as AD, PD, and strokes, which are the most burdensome diseases, globally and socially, is highly desired. Although gene therapy has great promise for the treatment of those neurological disorders, the delivery of therapeutic agents to the brain is the biggest hurdle. Thus, the development of brain-targeted DDS is as important as or even important than the drug itself. When designing a rational brain-targeted system, we should consider the differences in BBB structures and functions between normal conditions and pathological conditions. As represented by Zolgensma, AAV-vectors are the most advanced gene delivery vector for gene therapy, however, there are several remaining problems and challenges such as tissue-tropism preservation among animal species, and neutralization antibodies against AAV. On the other hand, non-viral gene delivery systems are safer, more flexible, and a cost-effective route compared to viral vectors. Actually, non-viral vectors have been reported and they have potential for the treatment of CNS diseases such as strokes, cancer, and neurological disorders. Although studies for developing brain-targeted non-viral gene delivery systems are currently in progress, the efficiency of expression continues to be inferior to that of viral vectors. We do not know the precise level of expression of therapeutic genes needed to produce therapeutic effects on CNS diseases. The efficiency of non-viral vectors may need to be improved, but there is no need to compete with viral vectors as long as the expression efficiency is sufficient to exert a therapeutic effect. It is possible that we will need quantitative data regarding the amount of therapeutic gene expression that is needed to cure the diseases for the purpose. As a common point of view between viral and non-viral vectors, it should be considered that brain-targeted vectors can accumulate in tissues other than the brain. Finding more specific targets and/or preventing other tissue accumulation are thought to be strategies for developing more brain-selective carriers. In order to achieve this, we will need to understand the biological mechanisms in healthy/pathological conditions, the characteristics of delivery vectors and the modality of interaction between the vectors and the living body. Mechanistic studies and some screening may be required to accomplish this. Although the present state of CNS-targeted drug development is still in the initial stage, recent promising innovations and future persistent research may provide us with novel and efficient drugs for treating CNS disorders.

## Figures and Tables

**Figure 1 pharmaceutics-12-01216-f001:**
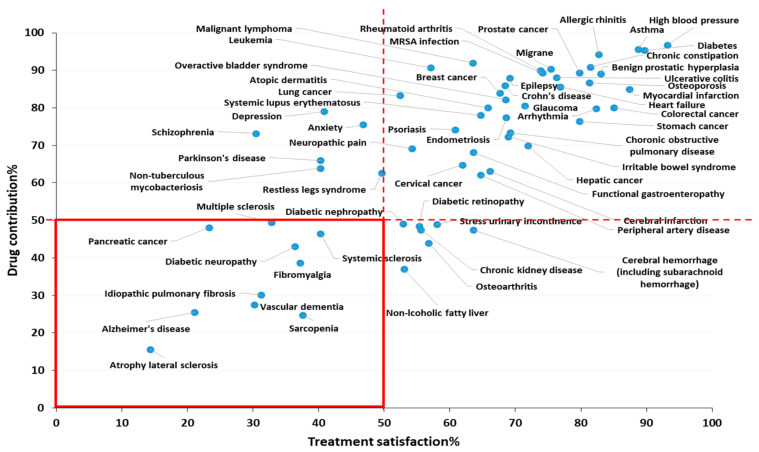
Treatment satisfaction and drug contribution. This figure is based on a medical need survey concerning medical satisfaction and drug contribution for physicians in Japan (2014~2019). Treatment satisfaction% was defined as the percentage of physicians who chose “Fully satisfied” and “Satisfied to some extent” from four options (1. Fully satisfied; 2. Satisfied to some extent; 3. Dissatisfied; 4. Not being treated) for the disease in the questionnaire. Drug contribution% was defined as the percentage of respondents who chose “Fully contributed” and “Contributed to some extent” of the four options (1. Fully contributed; 2. Contributed to some extent; 3. Not contributed; 4. No effective drugs) for each disease in questionnaire. This 2-axis dot plot based on a Japan questionnaire survey for physicians indicates the requirement of developing new drugs for the treatment of neurological disorders. There are 10 diseases with less than 50% of both treatment satisfaction and drug contribution, and 4 of them are neurological disorders including atrophy lateral sclerosis (ALS), Alzheimer’s disease (AD), multiple sclerosis (MS), and vascular dementia, figure is adopted with permission from [2], Japan Health Sciences Foundation, 2019.

**Figure 2 pharmaceutics-12-01216-f002:**
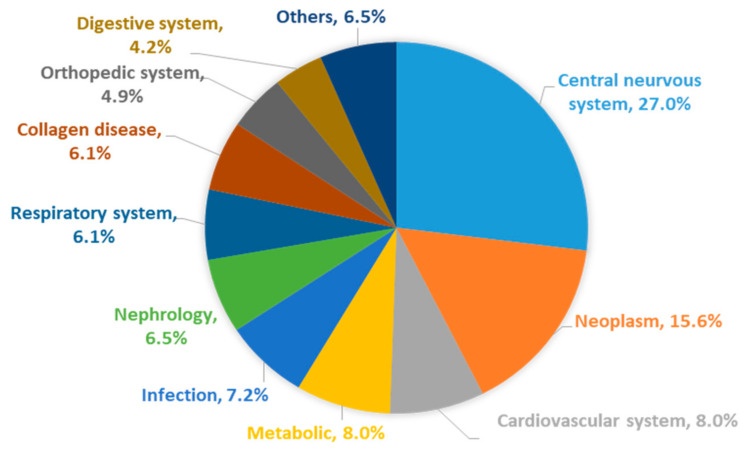
Diseases where there is an urgent need to develop new treatments and drugs. The pie chart is the result of a questionnaire concerning diseases that require the development of new drugs in Japan, figure is adopted with permission from [2], Japan Health Sciences Foundation, 2019. This figure was based on data from survey participants that were asked to list three diseases that they considered to be in urgently in need of new treatments and therapeutics, and then categorized those diseases into central nervous system (CNS), neoplasm, cardiovascular system, metabolic, etc. The result shows the diseases in most need of new drug development are CNS disorders.

**Figure 3 pharmaceutics-12-01216-f003:**
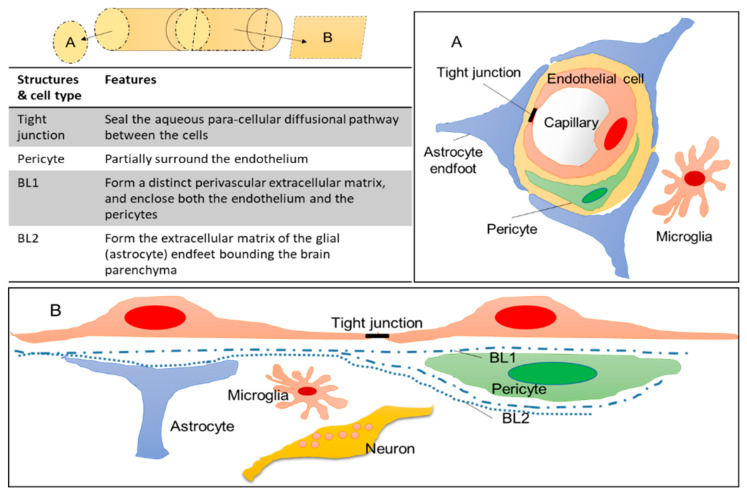
General structure of the BBB. Figure 3A,B show different cross-sectional views of brain blood vessels, respectively. The upper left picture shows the direction of cross-section of (**A**) (cut in round slices) and (**B**) (cut into squares) when the brain blood vessels are viewed as a cylinder. BL; basal lamina.

**Figure 4 pharmaceutics-12-01216-f004:**
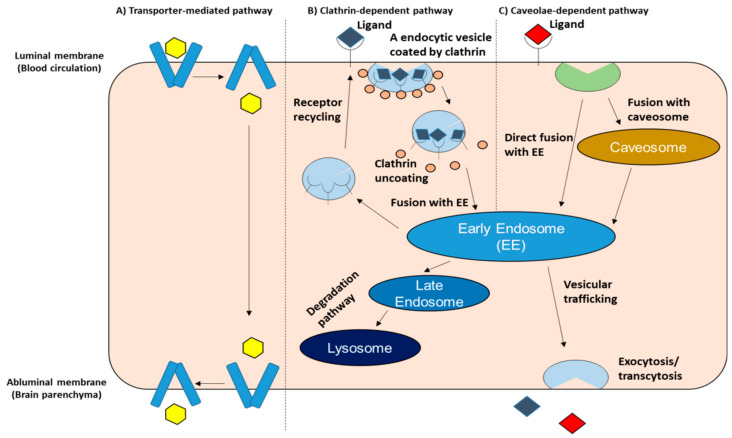
Main internalization pathways for receptors/transporters expressed on the BBB. (A) Transporter-mediated pathway: This pathway implies only an interaction between the ligand and the receptor. Conformation change is induced after ligand binding; (B) Clathrin-dependent pathway: This pathway requires the association of specific endocytic proteins, promoting the formation of clathrin-coated vesicles. The vesicles then dissociate from the membrane and undergo pH changes which promote the dissociation of clathrin and the ligand/receptor complex. The receptor is recycled; (C) The caveolae-dependent pathway: This pathway is regulated by the caveolin-1 and cavin proteins. The caveolae-dependent pathway has the ability to bypass lysosomal storage, which is different from the clathrin-dependent pathway. Little precise information if available regarding the BBB transcytosis mechanism, which is referred to as “Vesicular trafficking” in the figure, despite several reports indicating that transcytosis is regulated by Rab GTPases, which are a known group of molecules that control intracellular vesicle transport [82]. EE—Early Endosome.

**Figure 5 pharmaceutics-12-01216-f005:**
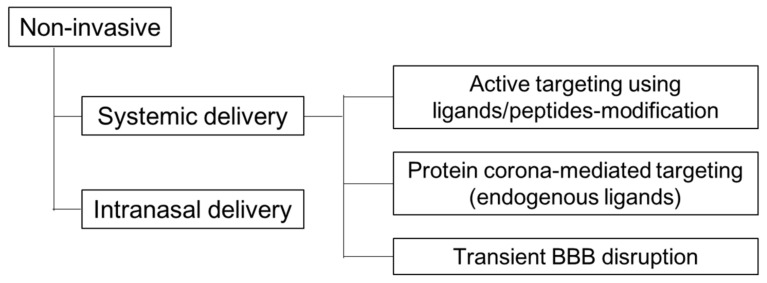
Non-viral strategies for brain targeting by non-invasive methods.

**Table 1 pharmaceutics-12-01216-t001:** Gene therapy products launched around the world.

Product (Company)	Vector	Delivery Gene	Target Disease	Approved Country (Year)	Reference
Gendicine (Shenzhen SiBiono GeneTech, Shenzhen, China)	Adenovirus (in vivo, IT)	Human wild-type p53	Head and neck squamous cell carcinoma	China (2003)	
Oncorine (Shanghai Sunway Biotech, Shanghai, China)	Oncolytic adenovirus (in vivo, IT)		Head and neck cancer	China (2006)	[26]
Rexin G (Epeius Biotechnology, Monrovia, CA, USA)	Retrovirus (in vivo, IV)	Mutant cyclin G1	Chemoresistant solid tumor	Philippines (2007)	
Neovagulgen (Human Stem Cell Institute, Moscow, Russia)	Plasmid (in vivo, IM)	Vascular endothelial growth factor (VEGF)	Peripheral artery disease	Russia (2011) Ukraine (2013)	
Glybera (Uniqure, Amsterdam, Netherlands) Discontinued	AAV1 (in vivo IM)	Lipoprotein lipase	Lipoprotein lipase deficiency	Europe (2012)	[27]
Imlygic (Amgen, Thousand Oaks, CA, USA)	Oncolytic HSV1 (in vivo, IT)	Granulocyte macrophage colony-stimulating factor (GM-CSF)	Malignant melanoma	America (2015) Europe (2015)	[28]
Strimveils (GSK, Brentford, United Kingdam)	Retrovirus (ex vivo)	Adenosine deaminase (ADA)	ADA deficiency	Europe (2016)	[29]
Zalmoxis (MolMed S.p.A, Milano, Italy)	Retrovirus (ex vivo)	Herpesvirus thymidine kinase	Graft versus host disease	Europe (2016)	
Kymriah (Novartis, Basel, Switzerland)	Retrovirus (ex vivo)	Chimeric antigen receptor (CAR) against CD19	Acute lymphoblastic leukemia	America (2017) Europe (2018) Japan (2019)	[30]
Yescarta (Kite Pharma, Santa Monica, CA, USA)	Retrovirus (ex vivo)	CAR against CD19	Large B cell lymphoma	America (2017) Europe (2018)	[31]
Luxturna (Spark Therapeutics, Philadelphia, PA, USA)	AAV2 (in vivo, SR)	RPE65	Retinal dystrophy	America (2017) Europe (2018)	
Zynteglo (Bluebird Bio, Cambridge, MA, USA)	Lentivirus (ex vivo)	Beta-globin	Beta thalassemia	Europe (2019)	[32]
Zolgensma (Novartis, Basel, Switzerland)	AAV9 (in vivo, IV)	SMN1	Spinal mascular atrophy (SMA)	America (2019) Europe (2019)	[25]
Collategene (Mitsubishi Tanabe Pharma Corporation, Osaka, Japan)	Plasmid (in vivo, IM)	HGF	Critical limb ischemia	Japan (2019)	

IT: intra-tumor, IV: intravenous, IM: intra-muscular, SR: sub-retinal.

**Table 2 pharmaceutics-12-01216-t002:** Gene therapy strategies and therapeutic effects for major CNS diseases.

Disease	Target Gene (Protein)	Mechanism	Reference
PD	AADC	Replace	[16]
	GAD	Replace	[17]
	GBA1	Replace	[33]
	SNCA	Silence	[34]
AD	APOE4	Silence/immunotherapy	[20]
	APOE2	Replace	[35]
	APP (amyloid-beta)	Silence	[22]
	MAPT (tau)	Silence/immunotherapy	[36]
	PQBP1	Replace	[23]
ALS	SOD1	Silence	[37]
	C9orf72	Silence	[38]
	TARDBP	Silence	[39]
SMA	SMN1	Replace	[24]

**Table 3 pharmaceutics-12-01216-t003:** Blood-brain barrier (BBB) breakdown and dysfunction in pathological conditions.

CNS Disease	The Type of BBB Breakdown and Dysfunction	Reference
AD	Down-regulation of glucose transporter 1 (GLUT1)	[201]
	Down-regulation of low density lipoprotein receptor (LDLR)	[203]
	Decreased expression and functionality of P-gp	[204]
	Low expression of tight junction (TJ) proteins	[205]
Multiple sclerosis (MS)	Abundance of chemokines and cytokines, leading to damage on TJ proteins	[206]
PD	Decrease in TJ protein expression, leaky BBB	[207]
	Abnormal distribution of GLUT1	[208]
Stroke	Up-regulation of GLUT1 Increased BBB permeability	[209]
		[210]
		[211]
Glioblastoma	Induction of blood-brain tumor barrier	[212]
	Overexpression of certain receptor-mediated transcytosis (RMT)	[213]

**Table 4 pharmaceutics-12-01216-t004:** CNS-targeted Adeno-associated virus (AAV) vectors developed so far.

Serotype	Relative Efficiency (vs. AAV9)	Notable Features Capsid Mutation Pattern	Reference
AAV9	1	Neonatal mice→neurons	[216]
		Adult mice→astrocytes	
		Used as the vector of Zolgensma	
AAV.GTX		AAV9 capsid VP1; Y446F and Y731F→inhibit ubiquitination	[217]
AAV-PHP.B	<40	AAV9 capsid; Q588-TLAVPFK-A589→AAV-PHP.B	[195]
AAV-PHP.eB		AAV-PHP.B capsid; A587D, Q588G→AAV-PHP.eB	
AAV.B1	~10	DNA shuffling, Less reactive to neutralizing antibodies than AAV9	[218]
AAV.AS	~15	AAV9 4 amino acids; S414N, G453D, K557E, T582I→AAV9.47	[219]
		AAV9.47 VP2; +19 alanine	

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
