# Peer review of "Current Status and Challenges Associated with CNS-Targeted Gene Delivery across the BBB"

_pharmaceutics, 2020, doi:10.3390/pharmaceutics12121216_

Round 1
Reviewer 1 Report
The manuscript entitled "Current Status and Challenges associated with CNS-Targeted Gene Delivery across the BBB” is a detailed review discussing aspects of drug delivery and particularly the delivery of gene therapy tools across the blood-brain barrier. I would say, that sometimes it is overly detailed, but perhaps that is necessary to make the points that the authors intend to make. Nevertheless the importance of brain targeted drugs is unquestionable in modern societies burdened by the detrimental effects of age on human cognitive function and that of aging related neurological disorders.
Critical comments:
-
The phrasing could be a bit more accurate regarding the BBB and its functions. It is probably better to phrase it something along the lines: The blood brain barrier is the interface that regulates transport between the circulation and neural tissue. The structural basis of the blood brain barrier is provided by the endothelial lining of the brain microvasculature. The brain microvascular endothelial cells are specialized and differ from peripheral capillaries: no fenestration, decreased pinocytosis and transcytosis, tight interendothelial junctions, high expression and asymmetric localisation of transporters and high metabolic activity. The development and maintenance of BBB function is dependant upon the concerted interaction of multiple cell types with the brain capillary endothelium, namely: pericytes, astrocytes, microglia and neurons.
-
The text that is referencing figure 4. Is a bit confusing about transcytosis being an internalization pathway. Also the figure show internalization only and the last step of transcytosis (e.g. abluminal transporters and exocytosis) should be added to the drawing.
-
When discussing the number of astrocytes (line 193), maybe cite DOI: 10.1002/cne.21974 and mention that the distribution of glia is not homogenous in the brain.
-
Regarding the use of the Gateway reflex as a modulator of BBB function it would be very educative for the reader if current methods for artificial neural activation were mentioned in a few sentences. Off the top of my head I only recall reading about deep brain stimulation using electrodes, but there could be less invasive methods as well. A single sentence discussion of the pros and cons of artificially activating neurons in a brain region would be great as well.
-
There are a few examples of brain targeting AAVs that could also be mentioned , that aim at targeting specific cell populations without sacrificing AAV cargo capacity for cell type specific promoter sequences:
Endothelial cell targeting: DOI: 10.15252/emmm.201506078
This one is a bit old, but the only example I know that is human astrocyte targeting by capsid selection: doi: 10.1038/mt.2009.184
There are of course other astrocyte targeting studies ass well: DOI https://doi.org/10.1038/s41434-019-0075-6
-
Manuscript would benefit from a proofreading by someone not yet familiar with the text. Not being a native english speaker myself I understand that cultural and language differences between the authors and me can render a perfectly clear sentence confusing for me, thus please forgive me if I am mistaken about some of the examples below.
Some examples of sentences that could be improved with proofreading:
Line 135: Either something is missing or the word “discuss is duplicated in: “conditions is needed. Thus, we discuss and discuss the BBB transporters, mechanisms of BBB “
Line 193: Astrocytes not „occupy”, but constitute a large portion of brain cells.
Line 434: “in order to survive the motor neurons “ a word is missing, maybe: “in order to improve the survival of motor neurons”
Line 473: “Since AAV is a naturally occurring virus, its infection can cause immune responses.” As natural origin of an antigen is not a prerequisite for immune response, the authors probably mean that as AAV infections occur naturally in humans, some patients can have preexisting immunity.
Line 552: The sentence contains dense information with little explanation about what the cited research is about, making it hard to understand: “FUS and MBs increased brain gene transfection and dopamine levels were restored by FUS-triggered MBs glial cell line-derived neurotrophic factor (GDNF) gene delivered treatment, which produced an abnormal rotation reduction in PD rats”
Reviewer 2 Report
Presented as a comprehensive review of the current status and challenges with respect to gene delivery into the CNS this review comes across more as a poorly focused and edited collection of references to a variety of in vitro, animal, and human trial based observations. It attempts to cover too much ancillary topics and ends up showing mechanistic bias by omission in some areas and losing the focus on the titled topic.
Specific areas of concern include:
1/ Poor focus. Too many subjects reviewed that are not particularly relevant to the title, for example inflammation, and poor development of the relationship between these subjects and the objective of the review, such as the role of neurovascular endothelial cell transporters.
2/ Poor cohesiveness. For example the section on "non-viral targeting by non-invasive methods" begins with a figure listing "invasive", "non-invasive" and "others" which are not all discussed and includes a section on "transient BBB disruption".
3/ Obscure language usage throughout including overuses of "we describe" and similar at the beginning of many sections;
4/ Over focus on mechanisms that are not that relevant to gene delivery and unproven hypotheses, such as the "Gateway reflex" rather than more solid mechanistic data concerning immune cell entry into the CNS tissues during pathological and protective immunity.
5/ Lack of editorial comment. There are strategies that show promise in human treatment and many others that have not moved out of in vitro or animal studies, some evidently dropped as the references are from years ago. It is important to make distinctions between these.
6/ Poor understanding of some of the studies cited, such as the use of rabies peptides for delivery across the BBB. Rabies virus does not cross the BBB but travels up axons and the relevance of rabies peptides is more to entry into this pathway.
7/ The inclusion of largely irrelevant material such as Table 1.
Reviewer 3 Report
In their submitted manuscript, Kimura and Harashima, summarize gene delivery to the CNS. In my opinion, there are several issues to be addressed that can potentially make this a stronger manuscript. They are as follows:
1) Figures 1 and 2 need a better explanation. For example, what does "Drug Contribution %" refer to. More description of these figures is needed.
2) For Figure 3, what does the cylinder at the top left of the figure refer to? The Figure legend needs to better describe the figure to walk the reader through the purpose of the figure.
3) In the Introduction, the authors refer to AAVs as “particularly heavily used to transduce the therapeutic genes”. Please explain why AAVs are preferentially used to transduce therapeutic genes as opposed to other viral vectors. This point is not clear in the text.
4) In Figure 4, it is unclear how the molecule is transcytosed if it traffics to an early endosome and then gets recycled back to luminal surface. How does it get from the early endosome to the brain?
5) Line 256-7 should read "NOT fully understood"
6) In the discussion of angiopep-2, the authors should mention the clinical trials and results of these trials using angiopep-2 as a brain delivery vehicle.
7) In section 4 Figure 5 the authors list various non-invasive strategies but only discuss a few of them.They should discuss all of the strategies in Figure 5 or remove Figure 5 and change the title of section 4 to better reflect what is actually discussed in the section.
8) Line 544: “The latter example is that precisely manipulating the
binding pattern of brain targeted apolipoproteins on the nanoparticle surface…”. How exactly is the binding pattern of apolipoproteins “precisely manipulated” and how is this different than the example in the preceding sentence about pre-coating with ApoE? It is unclear the way these sentences are written.
Reviewer 4 Report
In this work Kimura and Harashima perform a very exhaustive review of the state of the art in gene delivery methods to the brain. The authors have performed profound analysis of the relevant literature, including a large number of references, which is a good quality of this work, since provides important information for researchers that will use this work as a reference.
Having said that, I am afraid that the manuscript requires major revision for the use of English language (perhaps requiring the revision by a native speaker or a professional proofing service). In general one can comprehend what the authors intend to say, but some expressions are not used considering the connotations of the words used and could be misleading to non-experts in the field. As an example the authors refer always to the BBB as a rigid physiological barrier, which may be considered a mere physical barrier. Indeed it is more appropriate to consider the BBB as a functional structure, and it is more appropriate to refer to it as a “tight” physiological rather than a “rigid” physiological barrier. Another example is when the author define Japan as “one of the most aging countries”, since I guess they a referring to the Japanese population, but not the country itself. There are many examples on which language has to be improved.
Section 1 (Introduction) requires changes. In general there are many concepts mixed there and it is not well organized. It should be reduced. For example, there are several paragraphs and a figure talking about the BBB, and then in section 2 (BBB) they rewrite the same sentences. In my opinion, a couple of sentences in the introduction, “introducing” that there is a BBB and that represents a problem (without defining too much the BBB) is enough and then that paragraphs and the figure can be translated to section 2, where the BBB issues are discussed. The same can be said respect to AAVs and others. Introduction has to be modified to include only brief references to issues that are discussed in later sections and not include deep disclosure of information that is later repeated in other sections.
In lines 295-296 I believe that there is a mistake. The authors are talking about APOE bound polysorbate-80 coated nanoparticles, and later they said that this particles cross the BBB better than nanoparticles without polysorbate-80 coating. I believe the authors compare APOE and not APOE decorated lipid nanoparticles. Please revisit.
In the final section the authors discuss transient disruption of the BBB with ultra-sounds but actually it is also possible (and common) to disrupt the BBB transiently by osmotic shock (for example with mannitol) the authors should mention this option and add some references.
In the same sense, the manuscript ends up very abruptly, the authors present a figure/table (figure 5) where they mention other potential routes (like intranasal delivery, for example) and I believe that they should add, 1st a sentence or two at the end of the manuscript like: “There are other alternatives….like intranasal delivery or…..” (authors should include a couple or references for those other options) “…that are out of the scope of this work, but have also been efficient in allowing the delivery of…..” “the reader is pointed to the work of …..(add a couple of review references) for more information about these other options.”
Something like this will help to finish the section less abruptly.
Round 2
Reviewer 4 Report
The authors have introduce changes that have improved the quality of they work, and have satisfied all my previous concerns.